# Improving LLM Reasoning via Symbolic Inference over Logic Graphs

## Abstract

Large language models (LLMs) exhibit strong language understanding but remain limited in logical reasoning, particularly in multi-hop inference involving complex contextual dependencies. We propose **Graph-based Planned Reasoning (GPR)**, a neuro-symbolic framework that enhances LLM reasoning by organizing the process into structured stages. GPR builds a logic graph to capture fine-grained symbolic relations from natural language context, then leverages Planner to generate a goal-directed reasoning strategy. A dedicated Reasoner conducts stepwise symbolic inference along this plan, while Critic modules act as internal validators, checking and revising the logic graph and the final inference when necessary. This design enables GPR to perform faithful, interpretable reasoning while maintaining robustness against irrelevant or misleading information. Experiments across multiple logical reasoning benchmarks demonstrate that GPR consistently outperforms existing reasoning baselines and remains robust under noisy conditions.

## 1 Introduction

Large Language Models (LLMs) excel at natural language understanding and generation (Brown et al., 2020; Touvron et al., 2023; OpenAI, 2024a), but struggle with complex reasoning, especially multi-hop inference and contextual integration (Blair-Stanek et al., 2023). To address LLMs' reasoning limitations, prompting methods like Chain-of-Thought (CoT) Wei et al. (2022), Tree-of-Thought (ToT) Yao et al. (2023), and Graph-of-Thought (GoT) Besta et al. (2024) have been proposed. These guide LLMs to decompose problems into intermediate steps, improving transparency and accuracy.

Despite these advancements, LLMs remain brittle on reasoning tasks with many premises and multi-hop dependencies. They are particularly sensitive to input perturbations: irrelevant content distracts reasoning (Shi et al., 2023; Jones & Steinhardt, 2022), and even complete but disordered premises can impair performance (Chen et al., 2024).

This brittleness stands in sharp contrast to human reasoning, which remains robust in the face of irrelevant or disordered information. Such robustness stems from a cognitive strategy: before reasoning, humans filter distractions and organize information into structured forms that clarify dependencies and facilitate focused inference. Inspired by this, recent studies have explored injecting structural priors into LLMs. For example, InfoRE (Cheng et al., 2024) and COP (Liu et al., 2025) reorganize context into intermediate graph-based forms, such as mind maps, to surface logical relations and suppress noise. RoG (Luo et al., 2024) takes a different route by retrieving paths from external knowledge graphs to guide reasoning. While these approaches introduce structure, they share a limitation: their graphs are shallow, externally sourced, or ultimately converted back to plain text, limiting alignment with the context's internal logic and utility in symbolic inference.

To address these limitations, we propose Graph-based Planned Reasoning (GPR), a neuro-symbolic framework that constructs a context-specific logic graph with rich expressiveness. The graph captures fine-grained semantic units and logical dependencies from natural language input. Unlike prior work, GPR enables direct reasoning over the structured representation via symbolic rules, supporting faithful, interpretable, and robust multi-hop inference on logical reasoning tasks, rather than broad general-purpose or numerical reasoning.

The reasoning process in GPR is divided into three stages: **Logic graph construction**, which extracts and organizes logical relations from the context; **Reasoning plan formulation**, which outlines

a goal-directed inference path aligned with the graph; and **Reasoning with logic graph and plan**, where the logic graph and plan jointly support stepwise symbolic inference. Throughout the process, Critic modules assess the constructed graph and final answer, enabling the framework to detect and revise potential inconsistencies. An overview of the framework is shown in Figure 1.

We evaluate GPR on four logical reasoning benchmarks: *LogicBench* (Parmar et al., 2024), *FO-LIO* (Han et al., 2022), *ProofWriter* (Tafjord et al., 2021), and *AR-LSAT* (Zhong et al., 2022), comparing against prompting baselines including *standard few-shot*, *Chain-of-Thought (CoT)*, and *Tree-of-Thought (ToT)*, as well as recent neuro-symbolic approaches like *Logic-LM*(Pan et al., 2023) and *SymbCoT*(Xu et al., 2024). Across diverse LLMs, our framework consistently outperforms these baselines, often by substantial margins. Beyond accuracy, we further assess the robustness of GPR using a synthetic dataset, *IFI (Irrelevant Factual Interjection)*, created by injecting distractive content into *LogicBench*. GPR remains effective under noise, especially with dense or tail-positioned perturbations. Ablation studies validate the contribution of each component to performance.

In summary, our main contributions are:

- We propose Graph-based Planned Reasoning (GPR), a framework that constructs a context-specific logic graph capturing fine-grained logical relations for interpretable reasoning, with mechanisms that preserve consistency.

- GPR incorporates a goal-directed reasoning process that combines backward planning and symbolic inference over the graph, along with automatic answer verification to guarantee rigorous and transparent multi-hop reasoning.

- Experiments on four logical reasoning benchmarks demonstrate GPR's effectiveness in multi-hop reasoning. Additional evaluation on the IFI benchmark confirms its robustness to noisy context.

## 2 RELATED WORK

**Reasoning with LLMs** To enhance the reasoning capabilities of LLMs, various prompting strategies have been developed. Chain-of-Thought (CoT) Wei et al. (2022) encourages step-by-step reasoning, while Tree-of-Thought Yao et al. (2023) and Graph-of-Thought Besta et al. (2024) extend the process to tree and graph structures, aiming for better coverage and robustness. Further improvements aim to enhance the structure and reliability of the reasoning process. Self-consistency Wang et al. (2023b) improves answer robustness by sampling multiple reasoning paths, while Plan-and-Solve prompting Wang et al. (2023a) introduces an explicit two-stage framework that first generates a high-level plan and then executes it step by step.

Beyond purely neural strategies, a parallel line of research explores integrating symbolic reasoning into LLMs. Logic-LM Pan et al. (2023) and LINC Olausson et al. (2023) combine language models with external logic solvers for formal inference, but their effectiveness is limited by symbolic parsing quality and non-differentiable components. To overcome this, recent neuro-symbolic methods seek to more tightly couple symbolic structure with the neural reasoning process. SymbCoT Xu et al. (2024) enhances CoT by embedding first-order logic representations into intermediate reasoning steps, adopting a plan-then-solve architecture with explicit symbolic plans and a verifier for stepwise validation. In contrast, our method re-organizes the input context into a structured logic graph, enabling interpretable multi-hop reasoning grounded in symbolic dependencies.

**Structured Representation of Context** Structured context is vital for multi-step reasoning. Graph-based methods like AdaLoGN Li et al. (2022) use logic graphs and GNNs to model dependencies, while BASS Wu et al. (2021) and DGM Ouyang et al. (2021) convert text into semantic or discourse graphs to enhance comprehension. InfoRE Cheng et al. (2024) and COP Liu et al. (2025) reorganize context into mind maps to surface logical relations and filter noise. However, these are eventually linearized into plain text, limiting expressiveness and preventing direct inference. In contrast, our method builds a logic graph with typed nodes and edges, enabling richer representation and symbolic reasoning without reverting to unstructured text.

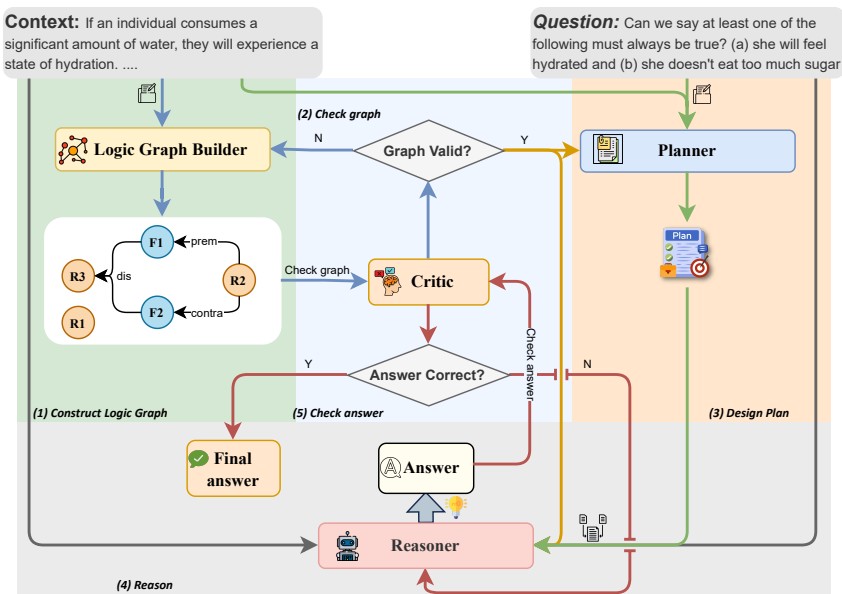

Figure 1: Overview of the workflow in our proposed Graph-based Planned Reasoning framework.

## 3 METHODOLOGY

In this section, we present Graph-based Planned Reasoning (GPR), a neuro-symbolic framework that enhances LLM reasoning by enabling symbolic inference over a logic graph, with planning guiding the process. As illustrated in Figure 1, GPR comprises four tightly integrated modules that are instantiated on a shared LLM backbone via structured interfaces: the Logic Graph Builder, which constructs a structured graph to model relational and logical structure from context so that free-form premises are converted into an unambiguous symbolic representation; the Critic, which performs post-hoc validation of the constructed graph and final answer to prevent errors; the Planner, which generates a goal-directed reasoning plan from the graph to mitigate drift that arises in unconstrained forward-only reasoning; and the Reasoner, which performs multi-hop inference by jointly leveraging the graph and the plan to ensure that every step follows symbolic rules rather than unsupported jumps. Although all roles use the same LLM, their behaviors are specialized by role-specific I/O schemas and rule constraints. The prompt configuration of our method is shown in Appendix I.3.

Concretely, all four modules interact through the same logic graph in a unified pipeline. The Builder constructs the graph, which serves as the shared structured representation. The Critic verifies it, after which the Planner derives a goal-oriented reasoning plan. The Reasoner executes the plan step by step on the graph, updating it with hypotheses and verified conclusions. Finally, the Critic checks the answer and coordinates corrections when needed.

Figure 2 illustrates the overall workflow of GPR through a representative example. The task is to determine whether Tom has a skeletal system, based on background knowledge about mammals. The reasoning process unfolds in five stages, involving four core components. For clarity, the Critic is not visualized in the figure, which focuses on illustrating the construction and use of the logic graph during planning and reasoning. First, the logic graph is constructed from the context, extracting facts and rules as typed nodes with symbolic edges (e.g., fact *F1* "Tom is a mammal" and rule *R1* "all mammals have vertebrae"). Second, the planner performs backward analysis from the goal ("Tom has a skeletal system") to identify a sequence of necessary inference steps, resulting in a reasoning plan $F1 \rightarrow R1 \rightarrow R2$, visualized as dashed orange edges. Finally, the reasoner executes symbolic inference along this plan: it derives the hypothesis *H1*, verifies it as *V1*, anchors it as fact *F3*, and continues inference until the final hypothesis *H2* is reached. This stepwise inference is traced by solid green edges. We elaborate on each stage in the subsections that follow.

## 3.1 TASK DEFINITION

Logical reasoning for multiple-choice question answering is the central focus of this task. The objective is to identify the correct answer option for a given question, leveraging a context that may require multi-step inference. Formally, each input sample is represented as a triple $(c, q, \mathcal{O})$, where $c$ denotes the context, $q$ is the question, and $\mathcal{O} = \{o_1, o_2, ..., o_k\}$ is a set of $k$ candidate options. The goal is to identify the correct answer $a^* \in \mathcal{O}$ using logical reasoning over $c$.

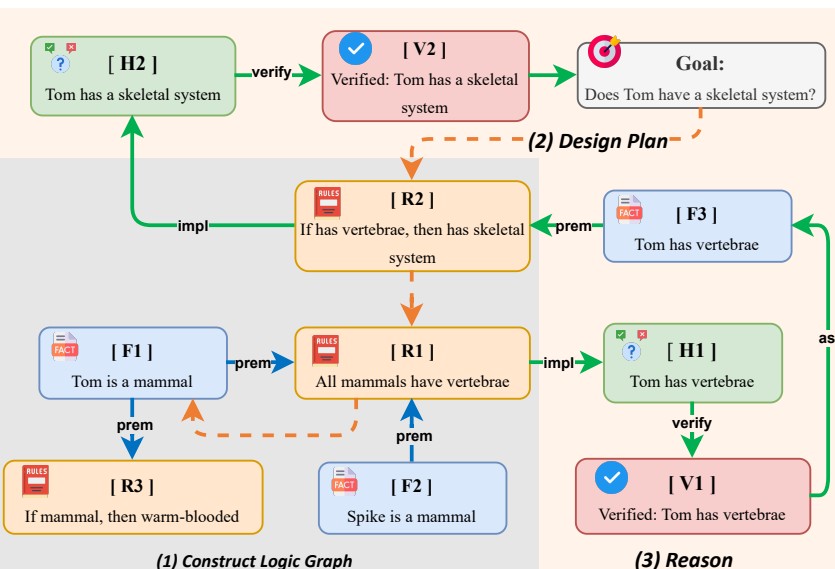

Figure 2: Goal-directed reasoning over a logic graph. The three stages are: (1) graph construction, (2) backward planning from the goal, and (3) symbolic reasoning. Blue edges indicate context relations, orange edges denote plans, and green edges trace reasoning. Node types: **F** = fact, **R** = rule, **H** = hypothesis, **V** = verified, **Goal** = question target.

## 3.2 LOGIC GRAPH

We introduce a graph-based representation, termed the *Logic Graph*, which abstracts the natural language context into a structured form to support interpretable reasoning. Formally, a logic graph is defined as $G = (V, E)$, where $V$ is a set of typed nodes and $E$ is a set of labeled edges that encode symbolic relations, such as implication and contradiction, among semantic units like facts, rules, and hypotheses. The logic graph serves as the foundation for downstream planning and inference. Its structure comprises two components: node representations and edge semantics.

As shown in Figure 2, the initial logic graph contains only fact and rule nodes. For example, the fact "Tom is a mammal" is node *F1*, while the rule "all mammals have vertebrae" is encoded as node *R1*. A directed *prem* edge connects *F1* to *R1*, indicating that the fact activates the rule.

| Node Type | Symbol | Description |
|---|---|---|
| Fact | *F* | Explicitly provided or anchored fact. Used in all reasoning edges. |
| Rule | *R* | Conditional logic rule activated by facts or other rules. |
| Hypothesis | *H* | Tentative conclusions from *impl*. Must be verified before reuse. |
| Verified | *V* | Result of *verify*. Optionally anchored as *F* or *R* via *as* for reuse, otherwise remains neutral. |

Table 1: Node types in the logic graph.

**Node Representation.** Each node $v \in V$ has a unique ID (e.g., F1, R3), natural language text, and one of four types: Fact (F) for assertions about individuals or existentially quantified statements, either explicitly provided in the context or subsequently established through verification and anchoring; Rule (R) for conditional statements, usually universally quantified; Hypothesis (H) for tentative

conclusions via *impl* edges; and Verified (V) for intermediate results via *verify* edges. Only F and R nodes appear in the initial logic graph; H and V are generated dynamically during reasoning. Verified nodes must be anchored as facts or rules via *as* edges before supporting further inference. Table 1 summarizes the four node types and their roles.

| Edge Type | Connection Pattern | Description |
|---|---|---|
| *prem* | $F \rightarrow R$ | A fact activates a rule. |
| *impl* | $R/F \rightarrow H$ | A rule or fact implies a hypothesis. |
| *verify* | $H \rightarrow V$ | Verifies a hypothesis into a verified node. |
| *as* | $V \rightarrow F$ or $R$ | Anchors a verified node as a fact or rule. |
| *con* | $\{F, R\}_n \Rightarrow V$ | Aggregates inputs via conjunction (AND). |
| *dis* | $\{F, R\}_n \Rightarrow V$ | Aggregates inputs via disjunction (OR). |
| *contra* | $F \leftrightarrow F / R \leftrightarrow R$ | Declares two facts or rules as mutually contradictory. |
| *equiv* | $F \leftrightarrow F / R \leftrightarrow R$ | Declares semantic equivalence between two nodes of the same type. |

Table 2: Edge types in the logic graph and their semantics. Notation: $\rightarrow$ indicates one-way dependency, $\Rightarrow$ indicates multi-to-one aggregation, and $\leftrightarrow$ indicates symmetric relations.

**Edge Semantics.** Edges in $E \subseteq V \times T \times V$ represent symbolic relations between nodes, where $T$ is a predefined set of edge types governing reasoning and composition. Each edge specifies a source, target, relation type (e.g., implication, verification), and optionally a rationale. Table 2 summarizes all edge types, their patterns, and semantic roles. More details are provided in Appendix A.

We distinguish two categories of edge types. Core inference edges support symbolic reasoning steps such as rule activation (*prem*), hypothesis generation (*impl*), verification (*verify*), and anchoring (*as*). Compositional and relational edges encode conjunction (*con*), disjunction (*dis*), contradiction (*contra*), and equivalence (*equiv*). Notably, Verified nodes can also result from composition via *con* or *dis* edges. In such cases, anchoring depends on the semantic roles of inputs: compositions of facts may be anchored as facts, rules as rules, while mixed inputs remain neutral until disambiguated.

### 3.3 GOAL-DIRECTED PLANNING ON GRAPH

To guide reasoning over the logic graph, we introduce a **Planner** that conducts backward analysis from the question to identify steps needed to reach the goal. Rather than executing inference directly, it outlines a high-level strategy by selecting relevant nodes and edges from the graph. Formally, given a logic graph $G = (V, E)$ and a question $q$, the Planner generates an ordered sequence $S = \{s_1, s_2, \ldots, s_m\}$, where each step $s_k = (V_k, E_k)$ specifies nodes and edges for that step. Appendix B shows a standalone visualization of the plan, and Appendix C details its implementation, including goal identification, path tracing, and completeness checks.

In our example (Figure 2), dashed orange edges illustrate the plan generated via backward analysis from the question "Does Tom have a skeletal system?" The planner identifies that answering requires applying rule *R2* ("if has vertebrae, then has skeletal system"), which depends on the intermediate conclusion "Tom has vertebrae." To establish this, it selects rule *R1* ("all mammals have vertebrae") and fact *F1* ("Tom is a mammal") as premises, yielding the reasoning plan $F1 \rightarrow R1 \rightarrow R2$.

### 3.4 REASONING WITH LOGIC GRAPH AND PLAN

Given a logic graph $G = \langle V, E \rangle$ and a reasoning plan $P = \{s_1, s_2, \ldots, s_k\}$, the LLM takes $G$, $P$, and symbolic rules as input, and performs step-by-step reasoning by interpreting the graph and applying symbolic inference rules until the goal is reached or no further inference possible. As shown in Figure 2, the solid green edges illustrate the reasoning. Starting from fact *F1* ("Tom is a mammal"), the model applies rule *R1* to derive hypothesis *H1* ("Tom has vertebrae"), verifies it as *V1*, anchors it as fact *F3*, and then uses rule *R2* to derive *H2* ("Tom has a skeletal system"). Each new conclusion is introduced as a hypothesis (*H*), verified as a node (*V*), and optionally anchored as a fact or rule (*F/R*) for reuse. The process ensures logical consistency and symbolic reasoning, while neural capacity resolves ambiguity and supports sound inference.

**Inference Rules.** We provide the LLM with propositional inference rules (e.g., Modus Ponens, Hypothetical Syllogism). See Appendix D for a list of these rules. Rules of inference for quantified statements, such as universal instantiation ($\forall x P(x) \vdash P(a)$), universal generalization ($P(a)$ for an arbitrary $a \vdash \forall x P(x)$), existential instantiation ($\exists x P(x) \vdash P(a)$ for some element $a$), and existential generalization ($P(a)$ for some element $a \vdash \exists x P(x)$), are not explicitly given. In a reasoning step, the LLM usually applies the combination of a propositional inference rule and one or more first-order inference rules. We give two representative examples below.

**Universal Modus Ponens:**
$$\forall x (P(x) \rightarrow Q(x)) \wedge P(a) \vdash Q(a) \tag{1}$$
In our example, rule *R1* ("all mammals have vertebrae") and fact *F1* ("Tom is a mammal") activate this pattern. The model applies Universal Modus Ponens to derive hypothesis *H1* ("Tom has vertebrae"), encodes it via an *impl* edge from *R1* to *H1*, verifies it as *V1*, and anchors *F3* via an *as* edge within the logic graph, thereby enabling its reuse in subsequent inference steps.

**Universal Hypothetical Syllogism:**
$$\forall x (P(x) \rightarrow Q(x)) \wedge \forall x (Q(x) \rightarrow R(x)) \vdash \forall x (P(x) \rightarrow R(x)) \tag{2}$$
Though not used in our example, this rule enables transitive inference. Given *R1*: "mammal $\rightarrow$ vertebrae" and *R2*: "vertebrae $\rightarrow$ skeletal system," the model derives *H'*: "mammal $\rightarrow$ skeletal system" via Universal Hypothetical Syllogism. In practice, we apply it stepwise to preserve traceability.

## 3.5 CRITIC

While the core reasoning is handled by the Builder, Planner, and Reasoner, GPR also incorporates a **Critic** that validates the logic graph's structure and meaning after construction, and inspects the final answer for correctness after reasoning.

**Graph Validation.** Before planning begins, the Critic verifies the logic graph in terms of definition consistency and semantic alignment. The former ensures that all node and edge types conform to the symbolic schema (e.g., implication edges must connect valid sources and hypotheses), while the latter checks whether the graph faithfully represents the core logical content of the original context. If issues are detected, the Critic initiates up to three refinement iterations with the Builder. This validation stage helps prevent downstream reasoning errors caused by flawed abstractions.

**Answer Validation.** After the Reasoner completes inference, the Critic verifies whether the final answer is logically supported by the verified graph. It checks that each step follows from valid premises, avoids contradictions, and makes no unsupported assumptions. If reasoning errors are detected, the Critic triggers a correction loop in which the Reasoner revises the inference path and retries reasoning. This stage is limited to two iterations.

## 4 EXPERIMENTS

**Models.** We evaluate SOTA LLMs, both proprietary and open-source. Specifically, we use GPT-3.5-Turbo (Ouyang et al., 2022), GPT-4-Turbo (OpenAI, 2024a), GPT-4o (OpenAI, 2024b), o1 (OpenAI, 2024c), and o3-mini (OpenAI, 2024d), all from OpenAI, and Grok-3 (xAI, 2025) from xAI. For open-source models, we evaluate DeepSeek-R1 (DeepSeek-AI et al., 2025), DeepSeek-V3 (DeepSeek-AI et al., 2024) and Qwen3-32B (Yang et al., 2025).[1]

**Datasets.** We evaluate our method on four logical reasoning benchmarks: LogicBench, ProofWriter, FOLIO, and AR-LSAT. These datasets encompass a diverse range of logical reasoning tasks, including first-order logic entailment, multi-hop deduction, and real-world analytical reasoning. Details about dataset construction, labeling schemes, and evaluation protocols are provided in Appendix E.

**Baselines.** We compare our method with five baselines, including prompting-based and neuro-symbolic approaches. *Standard* directly prompts the model for an answer without intermediate reasoning. *Chain-of-Thought (CoT)* (Wei et al., 2022) elicits step-by-step reasoning to decompose complex logic. *Tree-of-Thought (ToT)* (Yao et al., 2023) extends CoT by exploring multiple reasoning paths and selecting the best answer via search. *Logic-LM*(Pan et al., 2023) integrates LLMs with symbolic solvers by converting context into logic forms and verifying conclusions using an external

---

[1]We use GPT-4o-2024-08-06, DeepSeek-V3-0324 and Qwen3-32B-Instruct in our experiments.

prover. Following the original paper, if the generated program contains syntax errors, making it unexecutable by the solver, we adopt the model's CoT answer as a fallback. *SymbCoT* (Xu et al., 2024) augments CoT with logic plans and symbolic verification. All methods share the same in-context examples and use temperature 0. We report accuracy as the main evaluation metric for all tasks. Appendix I provides the full prompt configurations.

## 4.1 MAIN RESULTS

Table 3 shows the performance of our method (GPR) compared with five baselines: Standard prompting (Std), Chain-of-Thought (CoT), Tree-of-Thought (ToT), Logic-LM, and SymbCoT on four reasoning benchmarks. We highlight the following key observations:

| Method | GPT-3.5-turbo | GPT-4-Turbo | GPT-4o | o1 | o3-mini | Grok3 | DeepSeek-R1 | DeepSeek-V3 | Qwen3-32B |
|---|---|---|---|---|---|---|---|---|---|
| *LogicBench* | | | | | | | | | |
| Standard | 61.73 | 80.77 | 78.85 | 84.62 | 86.92 | 82.50 | 85.19 | 78.46 | 85.96 |
| CoT | 65.86 | 84.42 | 82.12 | 84.62 | 85.00 | 81.15 | 85.38 | 74.62 | 78.85 |
| ToT | 58.08 | 78.08 | 75.77 | 84.04 | 85.58 | 83.85 | 86.35 | 84.62 | 73.04 |
| Logic-LM | 40.77 | 67.31 | 59.81 | - | 83.27 | - | 81.35 | 71.35 | - |
| SymbCoT | 36.73 | 67.69 | 62.69 | - | 90.96 | - | 88.08 | 74.81 | 83.08 |
| GPR | **69.23** | **85.96** | **88.50** | **87.88** | **92.12** | **91.73** | **90.19** | **93.65** | **86.54** |
| *FOLIO* | | | | | | | | | |
| Standard | 45.09 | 59.80 | 67.86 | 77.45 | 79.90 | 77.94 | 80.88 | 79.41 | 66.81 |
| CoT | 57.35 | 67.65 | 72.14 | 77.94 | 77.94 | 75.98 | 79.90 | 73.53 | 70.10 |
| ToT | 52.45 | 70.10 | 72.55 | 79.41 | 80.88 | 80.39 | 80.88 | 77.94 | 73.04 |
| Logic-LM | 62.74 | 78.92 | 68.63 | 78.92 | 81.37 | 76.47 | 77.94 | 75.98 | - |
| SymbCoT | 57.84 | **83.33** | 75.98 | 81.37 | 84.80 | 82.84 | 84.80 | 79.90 | 77.94 |
| GPR | **63.24** | 81.86 | 78.92 | 83.82 | 88.24 | 86.76 | 85.78 | 85.29 | 79.41 |
| *ProofWriter* | | | | | | | | | |
| Standard | 35.50 | 40.67 | 56.33 | 87.00 | 84.00 | 90.83 | 89.00 | 73.00 | 55.33 |
| CoT | 49.17 | 64.33 | 67.33 | 88.17 | 84.83 | 84.17 | 88.00 | 73.67 | 64.67 |
| ToT | 39.33 | 70.83 | 67.83 | 86.50 | 82.67 | 88.83 | 86.17 | 83.33 | 77.83 |
| Logic-LM | 58.33 | 79.66 | 76.67 | 88.00 | 83.83 | 84.67 | 87.50 | 75.17 | - |
| SymbCoT | 59.03 | 82.50 | 80.00 | 88.33 | 85.67 | 89.67 | 88.67 | 86.17 | 79.33 |
| GPR | **61.33** | 83.83 | 82.16 | 91.00 | 87.17 | 95.33 | 93.17 | 88.00 | 80.50 |
| *AR-LSAT* | | | | | | | | | |
| Standard | 20.34 | 19.91 | 32.47 | 93.51 | 95.24 | 72.73 | 87.01 | 48.92 | 28.57 |
| CoT | 17.31 | 30.30 | 33.77 | 94.81 | 93.94 | 78.35 | 89.18 | 52.81 | 39.83 |
| ToT | 17.31 | 24.24 | 32.90 | 95.24 | 95.24 | 71.43 | 91.34 | 54.98 | 39.39 |
| Logic-LM | **26.41** | 43.04 | 33.33 | - | 94.37 | 79.65 | 90.04 | 55.84 | - |
| SymbCoT | 12.55 | 43.91 | 34.63 | - | 93.51 | 82.25 | 92.64 | 61.04 | 52.38 |
| GPR | 19.91 | **46.32** | 39.39 | 96.97 | 96.54 | 85.28 | 95.24 | 62.77 | 54.11 |

Table 3: Main results on four logical reasoning datasets. Each dataset includes results from six inference methods: Standard, CoT, ToT, Logic-LM, SymbCoT, and our method (GPR). **Bold** indicates the best result; underline indicates the second best.

1. GPR achieves consistently strong performance across all datasets and models. As shown in Table 3, GPR outperforms all baselines in most cases, covering both proprietary (e.g., GPT-4o, o1) and open-source models (e.g., DeepSeek-V3). On GPT-4o, it surpasses SymbCoT by 2.16% and CoT by 14.83% on ProofWriter; on GPT-4-Turbo, it leads CoT and ToT by 16.02% and 22.08% on AR-LSAT. Even on easier datasets, the margins remain significant.

2. GPR generalizes across base and reasoning models. It improves performance on weaker models (e.g., +11.15% over ToT on GPT-3.5, LogicBench) and remains effective on stronger models reasoning. For example, it boosts o3-mini from 84.00% to 87.17% and DeepSeek-R1 from 89.00% to 93.17% on ProofWriter. In contrast, baselines like CoT, ToT, or SymbCoT show inconsistent or even negative gains on stronger models, highlighting GPR's model-agnostic applicability.

3. GPR maintains high stability, while other methods show varied or fragile performance. Across benchmarks, second-best methods are inconsistent. SymbCoT performs well in some settings (e.g., AR-LSAT with Qwen3) but drops on others (e.g., LogicBench with GPT-3.5). ToT works well on DeepSeek-V3 but underperforms on weaker models. Logic-LM often fails to produce executable logic programs due to its strict requirement for well-formed logical forms, leading to fallback on CoT answers, as also noted in Pan et al. (2023). In contrast, GPR consistently ranks first or a close second, showing strong robustness across models and tasks.

## 4.2 ABLATION STUDIES

To assess each component's role in GPR, we conduct an ablation study on LogicBench across a range of models, including GPT-3.5-Turbo, GPT-4-Turbo, GPT-4o, Claude 3.7 Sonnet (Anthropic, 2024), and open-source LLMs. As shown in Table 4, we evaluate the impact of removing three core modules: *Planner*, *Logic Graph Builder*, and *Critic*. Specifically, **w/o Planner** removes planning and lets the Reasoner directly operate on the logic graph. **w/o Graph** removes the logic graph and performs reasoning solely based on a sequential plan from the planner, without graph-based structure. **w/o Critic** disables both graph-level consistency checks and final answer verification.

| Method | GPT-3.5-Turbo | GPT-4-Turbo | GPT-4o | o1 | o3-mini | DeepSeek-R1 | DeepSeek-V3 | Grok-3 | Claude 3.7 sonnet |
|---|---|---|---|---|---|---|---|---|---|
| **Full model** | 69.23 | 85.96 | 88.50 | 87.88 | 92.12 | 90.19 | 93.65 | 91.73 | 87.50 |
| w/o Graph | 62.88 | 84.04 | 85.38 | 85.19 | 87.88 | 83.65 | 91.73 | 87.88 | 84.04 |
| w/o Planner | 65.19 | 84.81 | 85.19 | 85.96 | 88.85 | 89.23 | 91.73 | 89.62 | 86.15 |
| w/o Critic | 67.88 | 85.19 | 87.50 | 86.92 | 91.35 | 89.42 | 93.46 | 90.77 | 86.54 |

Table 4: Ablation study of GPR on different LLMs by removing Logic Graph, Planner, and Critic modules.

Compared to the full model, all ablations degrade performance. The largest drop occurs without the Logic Graph, underscoring its foundational role in structured reasoning. Disabling the Critic weakens the system's ability to catch intermediate and final errors, leading to smaller declines. The absence of the Planner causes moderate loss, as unguided reasoning lacks goal-directed structure.

## 4.3 ROBUSTNESS TO IRRELEVANT INFORMATION

To evaluate the robustness of our method under real-world noise, we construct a dataset, **Irrelevant Factual Interjection (IFI)**, based on LogicBench. IFI injects distracting but factual information into the context, simulating scenarios where LLMs must identify key reasoning cues from a context augmented with irrelevant but plausible information.

Specifically, we interject segments randomly sampled from the WikiText dataset into the context along two dimensions: *insertion position* and *insertion density*. Insertion position is categorized into three levels: **Head**

| Position | Density | Std Acc (↓) | GPR (w/o Critic) Acc. (↓) | Acc. Gap | Drop Gap |
|---|---|---|---|---|---|
| Original (no IFI) | | 61.73 | 67.88 | +6.15 | – |
| Head | Low | 58.85 ↓2.88 | 65.19 ↓2.69 | + 6.34 | +0.19 |
| | Medium | 43.27 ↓18.46 | 60.19 ↓7.69 | + 16.92 | +10.77 |
| | High | 33.85 ↓27.88 | 56.92 ↓10.96 | +23.07 | +16.92 |
| Mid | Low | 56.35 ↓5.38 | 63.46 ↓4.42 | +7.11 | +0.96 |
| | Medium | 49.62 ↓12.11 | 57.50 ↓10.38 | +7.88 | +1.73 |
| | High | 35.96 ↓25.77 | 57.50 ↓10.38 | 21.54 | +15.39 |
| Tail | Low | 50.00 ↓11.73 | 59.81 ↓8.07 | +9.81 | +3.66 |
| | Medium | 47.69 ↓14.04 | 58.46 ↓9.42 | +10.77 | +4.62 |
| | High | 33.65 ↓28.08 | 61.35 ↓6.53 | +27.70 | +21.55 |

Table 5: **IFI robustness without Critic.** Each cell shows accuracy (%) / accuracy drop (↓) relative to the Original. **Acc. Gap** = accuracy difference between Ours and Std; **Drop Gap** = reduction in accuracy drop (smaller is better).

(at the beginning of the context), **Mid** (middle), and **Tail** (end). The insertion density defines how much distractor content is added, with three levels: **Low** (1× original length; 50% of final input), **Medium** (4× original length; 80%), and **High** (9×; 90%).

To assess the robustness of symbolic planning and reasoning, we disable the Critic, as filtering under noise may obscure core behavior. We compare **GPR** with a standard few-shot prompting baseline, implemented on *GPT-3.5-turbo*. As shown in Table 5, GPR yields better performance and less degradation across all settings. Notably, the advantage becomes more prominent under higher insertion densities, indicating robustness against irrelevant content. For instance, in the hardest setting (*Tail, High*), GPR retains an accuracy of 61.35%, achieving a 27.70% accuracy gain and a 21.55% drop reduction over the baseline. To further illustrate these gains, Appendix F presents a heatmap of margin over the baseline. The results underscore our method's ability to focus on logic-relevant structures and ignore distractions.

## 4.4 QUALITY OF LOGIC GRAPH

To assess whether the generated logic graph captures the context's logical structure, we design a semantic restoration task. Given only the graph as input, a strong LLM (DeepSeek-V3) is prompted to reconstruct the original natural language context. Since this is a single-pass decoding without Critic involvement, so the results directly reflect the intrinsic qual-

| Model | LogicBench | FOLIO | ProofWriter | AR-LSAT |
|---|---|---|---|---|
| GPT-3.5-Turbo | 60.00 | 67.65 | 49.17 | 63.20 |
| GPT-4-Turbo | 79.04 | 84.31 | 97.67 | 75.76 |
| GPT-4o | 77.69 | 88.24 | 95.00 | 90.04 |
| o1 | **89.81** | **91.67** | 97.00 | 91.77 |
| o3-mini | 87.50 | 89.22 | **99.17** | **93.07** |
| Grok3 | 85.77 | 82.35 | 99.00 | 65.80 |
| DeepSeek-R1 | 87.88 | 81.86 | 99.00 | 90.91 |
| DeepSeek-V3 | 85.19 | 82.35 | 98.83 | 92.64 |

Table 6: Semantic restoration accuracy (%) from logic graphs.

ity of the graph itself. Table 6 shows semantic restoration accuracy on four benchmarks. Stronger LLMs (e.g., GPT-4o, o1, o3-mini) achieve consistently higher scores, indicating that the graphs encode both semantic content and structural dependencies in a way that can be recovered by independent models.

A natural concern is that high restoration accuracy might only reflect semantic content preservation. To examine this, we analyze the correlation between restoration accuracy and reasoning performance. We find a significant positive correlation across models and datasets (Pearson's $r = 0.522$, $p = 0.0022$), suggesting that higher-quality graphs support better reasoning. Detailed dataset-wise results and scatter plots are in Appendix G.

### 4.5 ERROR ANALYSIS

To better comprehend where the errors in our GPR framework arise and how they are mitigated, an error analysis is conducted on the FOLIO dataset with GPT-3.5-turbo. All instances are examined and errors are categorized into three types capturing common failure modes:

- **Context Misunderstanding (CM)**: This occurs when the model misinterprets or overlooks crucial semantic cues in the premises, such as quantifiers, conditions, or negations. Such misunderstandings prevent the model from correctly capturing the intended meaning of the context.
- **Reasoning Chain Error (RCE)**: The model fails to construct a coherent multi-step reasoning process, either by skipping necessary intermediate steps, following an incorrect order, or reaching a conclusion without proper justification. These errors reflect difficulties in maintaining logical consistency across multiple inference steps.
- **Unsupported Inference (UI)**: This occurs when

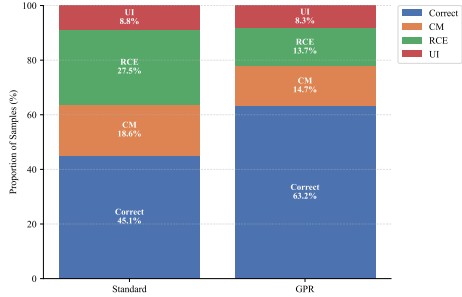

Figure 3: Distribution of error types on FOLIO for Standard baseline and GPR.

the model draws conclusions that are not justified by the premises, typically by introducing assumptions that are irrelevant, unverifiable, or contradictory to the given context.

When reasoning over FOLIO, all three error categories are observed in both methods. As illustrated in Fig. 3, Standard prompting is particularly prone to reasoning chain errors and also suffers from contextual misunderstandings. In contrast, GPR not only substantially reduces reasoning chain errors but also lowers contextual misunderstandings. These results indicate that our framework improves consistency in multi-step reasoning and accuracy of contextual interpretation, achieved by structuring premises into a logic graph and guiding inference through goal-directed planning.

### 5 CONCLUSION

In this paper, we propose a neuro-symbolic framework that enhances the reasoning capabilities of LLMs via logic graph construction and symbolic reasoning. Unlike previous approaches that reason directly over unstructured text, our method structures the context as a logic graph, highlighting logical relationships and multi-hop dependencies, and thus enables graph-based symbolic reasoning. To further improve reliability, we introduce a Critic that checks the logic graph and verifies the

final answer, helping prevent faulty abstractions and unsupported conclusions. These structural and verification mechanisms together enable more faithful and robust inference, even under noisy conditions. Empirical results on four benchmarks, along with evaluations under contexts augmented with irrelevant information, demonstrate consistent performance gains and robustness.

While effective, the framework may introduce structural redundancy in complex scenarios, where semantically similar facts or rules lead to repetitive patterns in the logic graph. Future work will explore graph compression techniques, such as node merging or hypergraph representations, to reduce redundancy while preserving logical completeness and improving inference efficiency.

## LIMITATIONS

While effective on a broad range of logical reasoning tasks, the framework is designed for scenarios where premise relations can be organized into a structured logic graph. Tasks dominated by numerical computation, heuristic search, or weakly structured linguistic context fall outside this intended scope. Extending the framework to handle such domains is a natural direction for future work.

## REPRODUCIBILITY STATEMENT

We describe all model settings, datasets, and evaluation protocols in the main text. An anonymous implementation is available at `https://anonymous.4open.science/r/ GPR-Reasoning-E35C`.

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

## A  EDGE TYPE DESCRIPTIONS

The logical edge types used in the logic graph are elaborated below:

**Core inference edges.**

- *prem* — Connects a fact ($F$) to a rule node ($R$); a rule is activated once one of its premises is satisfied. [2]
- *impl* — Links a rule or fact ($R/F$) to a hypothesis ($H$), meaning the hypothesis is logically entailed by the source.
- *verify* — Maps a hypothesis node to a verified node, recording the validation of an inferred statement. The resulting verified node remains untyped until explicitly anchored.
- *as* — Anchors a verified node ($V$) as either a fact or a rule ($F$ or $R$) so it can be reused in further inference.

**Compositional and relational edges.**

- *con* — Merges a set of facts or rules $\{F, R\}_n$ into a verified node ($V$) under conjunctive (AND) semantics. If all inputs are facts, the result is typically anchored as a fact. If all are rules, it may represent a compound rule and be anchored as a rule. For mixed inputs, the verified node remains unanchored and must be explicitly resolved before reuse.
- *dis* — Similar to *con*, but aggregates inputs via disjunctive (OR) semantics. Anchoring follows the same convention: pure facts lead to a fact, pure rules to a rule, and mixed inputs require disambiguation before downstream usage.
- *contra* — Bidirectionally marks two facts (or two rules) as mutually contradictory; they cannot both hold in the same reasoning branch.
- *equiv* — Bidirectionally marks two facts (or two rules) as semantically equivalent.

Together, these edge types enable explicit, verifiable neuro-symbolic inference over the logic graph.

## B  SUPPLEMENTARY VISUALIZATION OF REASONING PLAN

Figure 4 provides a standalone visualization of the goal-directed reasoning plan described in Section 3.3. It shows how the planner identifies a sequence of intermediate nodes and supporting edges that logically connect the goal statement to relevant rules and facts in the logic graph. This complements the integrated view in Figure 2.

## C  PLANNER CONSTRUCTION DETAILS

The Planner constructs the reasoning plan through the following steps:

1. **Goal Identification:** The Planner first determines the reasoning goal from $q$, such as verifying whether a disjunction $a \vee b$ holds, or whether a condition $c$ can be entailed from the context.

2. **Case Structuring (if applicable):** If the question involves disjunctive premises or alternative conditions (e.g., $F_1 \vee F_2$), the Planner decomposes reasoning into separate cases $C_1, C_2, \ldots$, each corresponding to one possible assumption path.

---

[2] If a verified node has been anchored as a fact ($V \twoheadrightarrow F$), it is treated as $F$ and can also serve as the source of a *prem* edge.

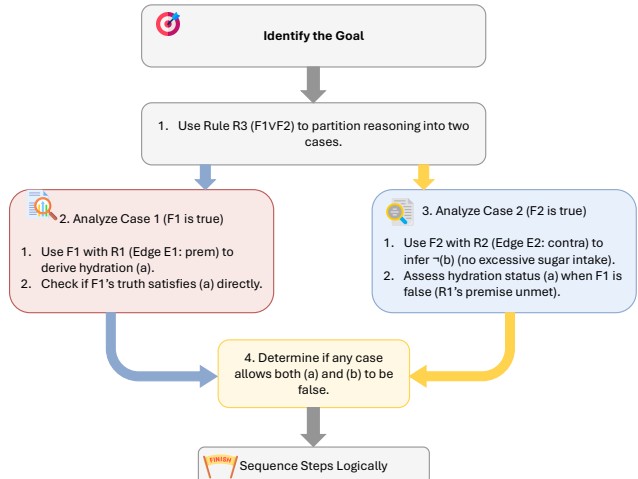

Figure 4: Supplementary visualization of a goal-directed reasoning plan over the logic graph.

3. **Graph-Based Path Construction:** For each case or directly from the question, the Planner outlines potential reasoning paths by traversing relevant nodes and edges in $G$. These paths reflect how existing premises, intermediate conditions, and rule nodes could be organized to approach the goal. The Planner does not confirm the truth of any step at this stage but instead identifies a logically coherent sequence.

4. **Completeness Check:** The Planner examines whether any path allows the query to be conclusively answered or refuted. If uncertainty remains, it ensures that all possibilities have been covered logically.

5. **Plan Assembly:** Finally, the steps $s$ are ordered into a coherent plan, ensuring that each inference builds on previous ones.

## D PROPOSITIONAL INFERENCE RULES

This appendix lists the core propositional inference rules summarized in Table 7. They serve as transformation schemata applied to the logic graph, while quantifier-related rules (e.g., universal instantiation) are handled implicitly during reasoning steps.

## E DATASET DETAILS

**LogicBench**  LogicBench (Parmar et al., 2024) is a recently proposed benchmark that evaluates symbolic reasoning through multiple formal logic tasks. We focus on its First-Order Logic subtask, where each instance takes the form of a Binary Question Answering (BQA) problem. Given a set of premises and a hypothesis, the task is to determine whether the hypothesis logically follows, with labels being *True* or *False*. We use the full test set of 520 examples.

**ProofWriter**  ProofWriter (Tafjord et al., 2021) contains synthetic yet linguistically natural problems requiring deductive reasoning under the open-world assumption. Each example consists of a set of premises and a goal statement, and the model must determine whether the goal is logically provable, disprovable, or undecidable. We adopt the depth-5 subset, which requires the most complex reasoning chains. To maintain label balance and reduce computational cost, we sample 600 test instances.

**FOLIO**  FOLIO (Han et al., 2022) is a challenging testbed for evaluating first-order logic reasoning in realistic contexts. Unlike synthetic datasets, FOLIO consists of human-authored examples using natural and diverse linguistic expressions. Each instance involves reasoning over a set of textual

| Rule | Inference Form |
|------|----------------|
| Modus Ponens | $(P \to Q),\ P \vdash Q$ |
| Modus Tollens | $(P \to Q),\ \neg Q \vdash \neg P$ |
| Hypothetical Syllogism | $(P \to Q),\ (Q \to R) \vdash P \to R$ |
| Transposition | $(P \to Q) \vdash (\neg Q \to \neg P)$ |
| Biconditional Intro. | $(P \to Q),\ (Q \to P) \vdash P \leftrightarrow Q$ |
| Biconditional Elim. | $(P \leftrightarrow Q) \vdash (P \to Q),\ (Q \to P)$ |
| Conjunction Intro. | $P,\ Q \vdash P \wedge Q$ |
| Conjunction Elim. | $(P \wedge Q) \vdash P \qquad (P \wedge Q) \vdash Q$ |
| Disjunction Intro. | $P \vdash P \vee Q \qquad Q \vdash P \vee Q$ |
| Disjunction Elim. | $(P \vee Q),\ (P \to R),\ (Q \to R) \vdash R$ |
| Disjunctive Syllogism | $(P \vee Q),\ \neg P \vdash Q \qquad (P \vee Q),\ \neg Q \vdash P$ |
| Double Negation | $\neg\neg P \vdash P$ |
| Negation Intro. (Reductio) | $(P \to \bot) \vdash \neg P$ |

Table 7: Core propositional inference rules (used as transformation schemata on the logic graph).

premises to judge whether a given hypothesis is true, false, or unknown. We use the entire 204-example test set for evaluation.

**AR-LSAT** AR-LSAT (Zhong et al., 2022) collects real-world analytical reasoning questions from historical LSAT exams, targeting deductive reasoning over abstract constraints. Each instance is a multiple-choice question involving spatial, logical, or rule-based constraints. We evaluate on the full test set of 231 examples.

## F ROBUSTNESS HEATMAP

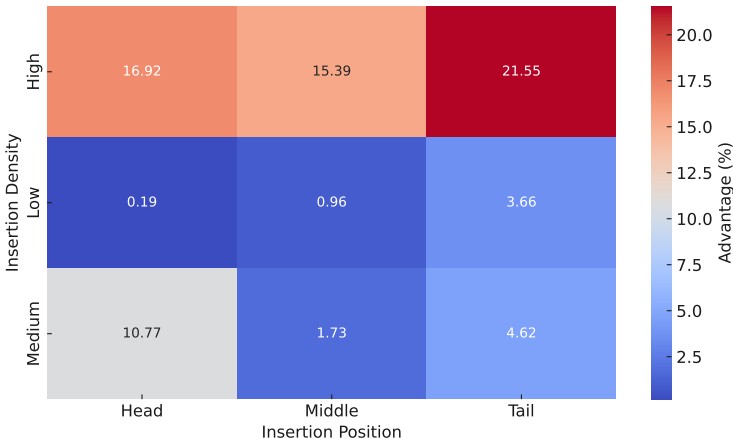

Figure 5: Performance advantage (%) of our method over *Standard* on the IFI dataset under varying insertion positions and densities.

## G CORRELATION BETWEEN SEMANTIC RESTORATION AND REASONING ACCURACY

**Setup.** We compute the Pearson correlation between reasoning accuracy and semantic restoration accuracy across models and datasets to test whether higher-quality logic graphs (as measured by

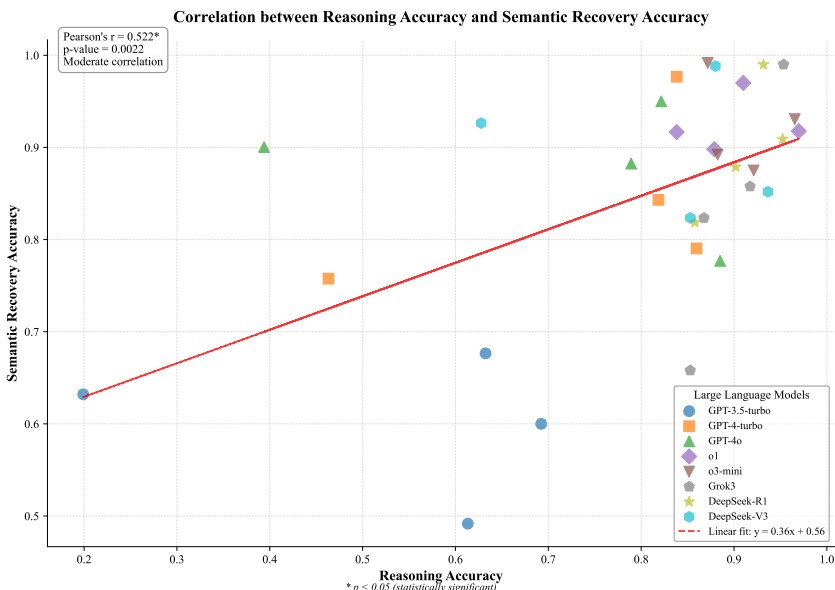

Figure 6: **Overall correlation between reasoning accuracy and semantic restoration accuracy.** Each point corresponds to a (model, dataset) pair. Markers denote different LLMs.

restoration) align with stronger downstream reasoning. For the overall analysis, each point is a (model, dataset) pair; for the dataset-wise analysis, each point is a model instance evaluated on that dataset. We report the correlation coefficient $r$ and two-sided $p$-value. A simple least-squares linear fit is overlaid in the overall plot for visualization.

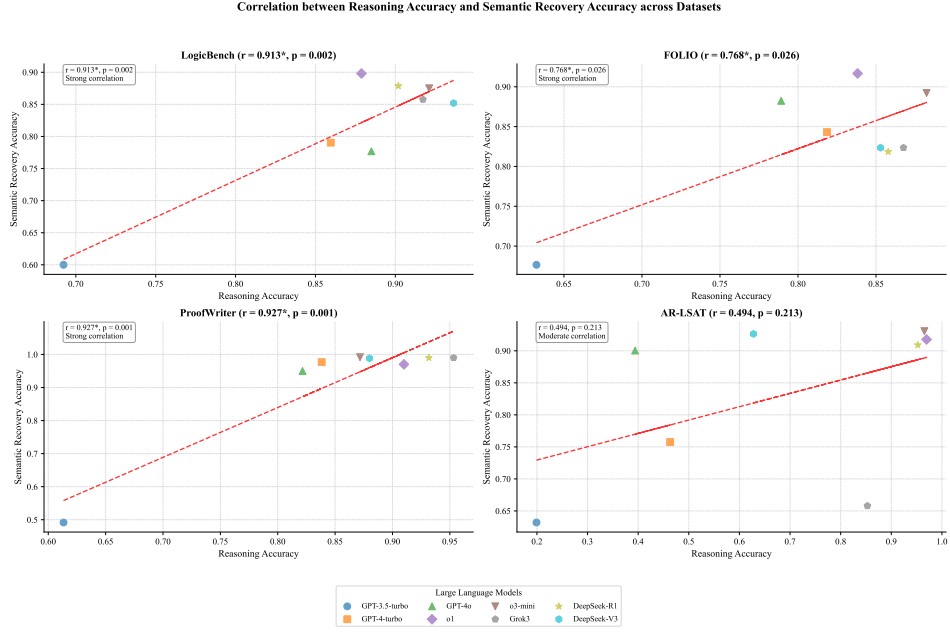

Figure 7: **Dataset-wise correlations.** Separate scatter plots for LOGICBENCH, FOLIO, PROOFWRITER, and AR-LSAT. Strong, significant correlations are observed in the first three datasets; AR-LSAT shows a moderate, non-significant trend.

**Results.** Overall, we observe a statistically significant positive correlation ($r = 0.522$, $p = 0.0022$), indicating that improvements in semantic restoration are associated with higher reasoning accuracy.

By dataset, correlations are strong and significant on LOGICBENCH ($r = 0.913$, $p = 0.002$), FO-LIO ($r = 0.768$, $p = 0.026$), and PROOFWRITER($r = 0.927$, $p = 0.001$), and moderate but not statistically significant on AR-LSAT ($r = 0.494$, $p = 0.213$).

**Discussion.** AR-LSAT shows a weaker, non-significant trend, likely due to its heterogeneous constraints and discrete option structure, where small restoration differences may not linearly translate to accuracy. Nevertheless, the consistent positive trends elsewhere substantiate the validity of semantic restoration as an informative, task-agnostic proxy for logic-graph quality.

# H CASE STUDY: END-TO-END GPR EXAMPLE

### Context and Question

**Context:** All growth companies' stocks are volatile. If the stock price is volatile, then it is not suitable for a retirement fund. Some companies' stocks are growth companies' stocks. All mature companies' stocks are suitable for a retirement fund. KO is a mature company's stock.
**Question:** Based on the above information, is the following statement true, false, or uncertain? KO is a company stock.

### Logic Graph Construction

**Nodes:**

- F1 (Fact): "All growth-company stocks are volatile."
- F2 (Fact): "If a stock is volatile, then it is not suitable for a retirement fund."
- F3 (Fact): "Some companies' stocks are growth-company stocks."
- F4 (Fact): "All mature-company stocks are suitable for a retirement fund."
- F5 (Fact): "KO is a mature-company stock."
- R1 (Rule): "If a stock belongs to a growth company, then it is volatile."
- R2 (Rule): "If a stock is volatile, then it is not suitable for a retirement fund."
- R3 (Rule): "If a stock belongs to a mature company, then it is suitable for a retirement fund."

**Edges:**

- E1: F1 → R1 *(prem)*
- E2: F2 → R2 *(prem)*
- E3: F4 → R3 *(prem)*
- E4: F5 → R3 *(prem)*
- E5: F3 → R1 *(prem)*

### Goal-Directed Reasoning Plan

**Goal:** Determine whether $G$: "KO is a company stock." can be inferred.
**Plan Outline:**

- Step 1: Use F5 together with R3 to derive whether KO is suitable for a retirement fund.
- Step 2: Verify the result and anchor it as a new fact for reuse.
- Step 3: Search the graph for any rule or fact that directly or indirectly connects "mature-company stock" or "suitable" to "company stock".
- Step 4: Examine general chains involving growth-company stocks and volatility, noting that they do not refer to KO.

### Step-wise Reasoning and Answer

**Step 1: Derive KO's suitability using explicit inference rules.**
From F5 ("KO is a mature-company stock") and R3 ("If a stock belongs to a mature company, then it is suitable for a retirement fund"), the Reasoner applies **Universal Modus Ponens**:

$$\forall x(\text{mature}(x) \rightarrow \text{suitable}(x)), \quad \text{mature}(KO) \vdash \text{suitable}(KO).$$

This yields:

- H1: "KO is suitable for a retirement fund." *(impl via Universal Modus Ponens)*

- V1: "KO is suitable for a retirement fund." *(verify)*

- F6: "KO is suitable for a retirement fund." *(as)*

**Step 2: Attempt to infer the goal $G$ using rule-based backward search.**
The Reasoner inspects all rules and fact–rule edges activated via `prem`, checking whether any rule can support an **impl** edge toward the hypothesis:

$$H_G : \text{"KO is a company stock"}.$$

The logic graph includes:

- growth-company stock $\rightarrow$ volatile $\rightarrow$ not suitable (F1, F3, R1, R2),

- mature-company stock $\rightarrow$ suitable (F4, F5, R3).

However, none of these rules have consequents that mention "company stock", and no rule's antecedent matches F5 or F6 in a way that could activate a chain leading to "company stock".
Thus, no impl edge to $H_G$ can be formed; i.e., there is no applicable inference rule (such as Modus Ponens, Hypothetical Syllogism, or Disjunction Elimination) that can produce the target hypothesis.

**Step 3: Examine possible indirect derivations.**
The Reasoner uses general rule chaining (Hypothetical Syllogism, Conjunction Introduction) on:

- F1 + R1 $\rightarrow$ (growth-stock $\rightarrow$ volatile)

- R1 + R2 $\rightarrow$ (growth-stock $\rightarrow$ not suitable)

These applications rely on:

- **Hypothetical Syllogism**,

- **Conjunction Introduction / Elimination** (Rules 7–8).

But all such derived hypotheses concern generic "some companies' stocks", not KO, and none introduce the predicate "company stock". Therefore, no new H node concerning KO can be generated.

**Conclusion.**
No sequence of symbolic inference steps—using Modus Ponens, Hypothetical Syllogism, Disjunction Elimination, or any other rule yields the conclusion "KO is a company stock." Likewise, the negation cannot be derived.

**Final Answer:** `C` (uncertain)

# I  PROMPT EXAMPLES

## I.1  CHAIN-OF-THOUGHT (COT)

### Baseline: Chain-of-Thought Prompt

**Task Description:** Given a problem statement as context, the task is to answer a logical reasoning question.
**Input Format:**

- *Context:* A list of facts and conditional rules in natural language.

- *Question:* A true/false/uncertain query about the context.

- *Options:* A) True    B) False    C) Uncertain

**Output Format:**

- *Reason*: Step-by-step natural language reasoning.

- *Answer*: A / B / C

**Example:**
*Context:* The dog sees the rabbit. If something sees the rabbit then the rabbit is cold. $\cdots$
*Question:* Is the rabbit cold?

> *Reason:* The dog sees the rabbit. The rule says if something sees the rabbit then the rabbit is cold. So the rabbit is cold.
> *Answer:* A

## I.2 TREE-OF-THOUGHT (ToT)

---

**Baseline: Tree-of-Thought Prompt (ToT)**

**Task Description:** Imagine three different experts are answering this question. Each expert writes down one step of their reasoning, shares it, and then proceeds to the next step. If any expert realizes they are wrong, they exit the process.

**Input Format:**

- *Context:* Natural language facts and rules.

- *Question:* Logical statement to verify.

- *Options:* A) True    B) False    C) Uncertain

**Output Format:**

- *Reason*: Multi-expert step-by-step reasoning, e.g.,

```
Expert 1: Step 1 ...
Expert 2: Step 1 ...
Expert 3: Step 1 ...
...
Expert 1: Step N ...
```

- *Answer*: A / B / C

**Example:**

*Context:* If people are vegetarian, then they are conscious about environment or health. · · ·
*Question:* If Jeremy has a busy schedule without time to cook, then he does not enjoy hamburgers and steaks?
*Reason:*

- Expert 1 deduces: busy schedule → fast food

- Expert 2: fast food → not environmentally conscious

- Expert 3: not conscious → not vegetarian → eats meat

- · · ·

- All experts agree: Jeremy enjoys burgers and steaks ⇒ original statement is false.

*Answer:* B

---

## I.3 GRAPH-BASED PLANNED REASONING (GPR)

---

### Step-1: Constructing the Logic Graph

**Task Description:** Given a natural language *context* and a corresponding *question*, this step extracts the underlying logical structure and organizes it into a symbolic graph. The graph captures the semantic dependencies and logical entailments among propositions.

**Input Format:**
- *Context:* Natural language paragraph expressing facts, rules, or constraints.
- *Question:* A query requiring logical inference.

**Output Format:**
- Enclosed in `<Graph>` tags.
- List of nodes: ID, text, and type.
- List of edges: ID, source, target, type, and rationale.

**Node Types:**
- *Fact:* Atomic statements directly from the context (e.g., "The sky is blue.").
- *Rule:* General implications with condition and consequence.
- *Hypothesis:* Derived statements requiring validation.
- *Verified:* Hypotheses confirmed and anchored into the graph.

**Edge Types:**
- *prem*: Links a fact to a rule as its premise.
- *impl*: Indicates a rule or fact implies a hypothesis.
- *verify*: Confirms a hypothesis.
- *as*: Anchors a verified node as a fact or rule.
- *con/dis/contra/equiv*: Represent conjunction, disjunction, contradiction, or equivalence.

**Important Notes:**
- Only include propositions explicitly stated in the context.
- Do not treat the question as part of the graph.
- Do not introduce new inferences or assumptions at this step.

---

### Step-2: Validating Logic Graph with Critic

**Task Description:** Before reasoning begins, the Critic validates the constructed logic graph to ensure symbolic and semantic correctness.

**Validation Dimensions:**
- **Definition Consistency:** All nodes and edges must follow the schema. Examples: `impl` edges connect rules/facts to hypotheses; `verify` edges link hypotheses to verified nodes; verified nodes must be anchored via `as` before reuse.
- **Semantic Alignment:** The symbolic graph must faithfully capture the logical relations expressed in the natural language context.

**Output Format:**
- Boolean value: `true` / `false`

---

### Step-3: Designing Goal-Directed Reasoning Plan

**Task Description:** Given a symbolic *logic graph*, this step generates a goal-directed reasoning plan that outlines how to approach the question through systematic symbolic inference.

**Node Types:**
- *F (Fact):* Explicit atomic information.
- *R (Rule):* Logical implication with premise $\Rightarrow$ conclusion.
- *H (Hypothesis):* Derived statement (requires validation).
- *V (Verified):* Confirmed intermediate result (must be anchored).

**Edge Types:**

---

- *prem*: F/V → R (premise activates a rule).
- *impl*: F/R/V → H (implication to hypothesis).
- *verify*: H → V (verify hypothesis).
- *as*: V → F/R (anchor verified node).
- *con/dis*: Multiple → H/V (conjunction / disjunction).
- *contra/equiv*: Bidirectional contradiction / equivalence.

**Planning Instructions:**

- Identify the goal from the question.
- Outline logical steps (1–5), referencing specific nodes/edges (e.g., "Use F1 and R2 to derive H3").
- Sequence steps coherently so that each builds upon previous ones.
- Stay neutral: do not commit to conclusions; only map logical routes.

**Output Format:**

```
<Plan>
1. Goal: Determine whether "[target statement]" can be inferred.
2. Steps:
   - Step 1: ...
   - Step 2: ...
3. Sequence logically.
</Plan>
```

## Step-4: Reasoning with Logic Graph and Rules

**Task Description:** Given a *logic graph* and the corresponding *goal-directed plan*, this step performs symbolic reasoning by applying logical inference rules over the structured graph. The objective is to derive the conclusion through step-wise deduction.

**Supported logical inference rules:**

- **Modus Ponens (MP):** $(P \rightarrow Q)$, $P \vdash Q$.
  Usage: If the graph contains a rule node for $P \rightarrow Q$ and a fact node for $P$, infer $Q$ as a hypothesis; record an `impl` edge from the triggering rule to $Q$, then `verify` and as-anchor $Q$ as a fact.
- **Hypothetical Syllogism (HS):** $(P \rightarrow Q)$, $(Q \rightarrow R) \vdash (P \rightarrow R)$.
  Usage: If two rule nodes encode $P \rightarrow Q$ and $Q \rightarrow R$, you may infer $(P \rightarrow R)$ as a rule-level hypothesis; `verify` it and as-anchor it as a new rule. When appropriate, prefer stepwise chaining $(P \Rightarrow Q \Rightarrow R)$ for traceability.
- ...

**Graph Elements:**

- *F (Fact):* Atomic, stated information.
- *R (Rule):* If-then logical constraints.
- *H (Hypothesis):* Derived statements.
- *V (Verified):* Confirmed intermediate results (must be anchored).

**Edge Types:**

- *prem*: premise activation $(F \rightarrow R)$
- *impl*: implication $(F/R \rightarrow H)$
- *verify*, *as*: verification and anchoring
- *con*, *dis*: conjunction / disjunction
- *contra*, *equiv*: contradiction / equivalence

**Output Format:**

```
Step-by-Step Explanation:
Step 1: Analyze F1 ...
Step 2: Apply R2 and R3 ...
Step 3: Contradiction detected between R1 and R3 ...
...
Final Answer: C
```

> **Step-5: Validating Final Answer with Critic**
>
> **Task Description:** After reasoning concludes, the Critic validates the final answer to ensure it is logically supported and free from errors.
> **Validation Dimensions:**
> - **Inference Validity:** All steps must follow from valid premises using permissible logic rules.
> - **No Unsupported Assumptions:** The answer must not rely on unstated or external knowledge.
> - **Contradiction Check:** Ensure no internal contradictions remain in the final graph state.
>
> **Output Format:**
> - Boolean value: `true` / `false`

## J   THE USE OF LARGE LANGUAGE MODELS (LLMS)

Large language models (LLMs) were used exclusively as assistive tools for grammar checking and minor wording adjustments during the preparation of this manuscript. They were not involved in research ideation, experimental design, data analysis, or interpretation of results. All scientific contributions are solely the responsibility of the authors.

