# OpenReview forum: "Improving LLM Reasoning via Symbolic Inference over Logic Graphs"
_ICLR.cc/2026/Conference — Submitted to ICLR 2026_

### Official Review · Reviewer_j6Rz · 2025-10-23

**Soundness:** 3
**Presentation:** 2
**Contribution:** 3
**Rating:** 6
**Confidence:** 3

**Summary:**

The paper proposes Graph-based Planned Reasoning (GPR), a framework comprising three main modules: Logic Graph Construction, Reasoning Plan Formulation, and Reasoning with Logic Graph and Plan. Extensive experiments across a wide range of benchmarks and diverse LLMs demonstrate that GPR achieves strong performance, maintains high stability, and exhibits robustness to irrelevant information.

**Strengths:**

- The proposed methods are novel and interesting.

- GPR achieves consistent empirical improvements over baseline methods across multiple benchmarks and model architectures.

- The experiments and analyses are comprehensive, and the availability of code enhances reproducibility.

**Weaknesses:**

- The methodology lacks motivation and design rationale. The paper would benefit from explaining why each design choice was made before introducing the method in detail.

- While each component is clearly described, the paper does not sufficiently explain how the components interact as a unified system. As a result, the overall workflow remains somewhat unclear.

- The error analysis is valuable, but the paper could be strengthened by discussing how future research might build upon these findings or what insights they offer for subsequent work.

**Questions:**

- What are the design motivations behind each component?

- How were the node and edge types defined?

- Why were only two symbolic rules chosen when many alternatives exist?

- Which components depend on LLMs, and which are implemented independently? (Related to the methodological clarity mentioned above.)

---

> ### Author Response · Authors · 2025-11-23
> **Response to Reviewer j6Rz**
>
> We thank the reviewer for the positive assessment of our work and for recognizing the novelty and empirical strength of the proposed framework. We now address the reviewer’s concerns in detail.
>
> > **W2:** While components are clearly described, the paper does not sufficiently explain how they interact as a unified system, leaving the overall workflow somewhat unclear.
>
> **A1:** We appreciate this comment. Although Figures 1 and 2 already illustrate the overall workflow, we agree that the text can more clearly articulate how the modules operate together. The Builder first constructs the logic graph, which becomes the shared structured representation for the entire pipeline. The Critic verifies this graph, after which the Planner uses it to derive a goal oriented reasoning plan. The Reasoner then executes the plan step by step on the same graph, updating it with hypotheses and verified conclusions. Finally, the Critic checks the answer and coordinates corrections if needed.
>
> > **W3:** The error analysis is valuable, but the paper could be strengthened by discussing how future research might build upon these findings or what insights they offer for subsequent work.
>
> **A2:** Thank you for the suggestion. The remaining errors in contextual interpretation, incomplete reasoning steps, and unsupported conclusions reflect fundamental limitations of LLMs in extracting and abstracting structured logical information. These findings point to a concrete direction for future research: improving the induction and abstraction quality of the logic graph so that downstream planning and reasoning operate on more accurate and verifiable structure. Possible approaches include supervised fine-tuning to strengthen graph construction, reinforcement learning to enforce structural correctness and consistency, and hybrid methods that combine neural extraction with lightweight symbolic checks. We will add this discussion to clarify how the observed errors motivate these avenues for future research.
>
> > **Q1:** What are the design motivations behind each component?
> >
> **A3:** Thank you for the suggestion. We will make the design motivations explicit.
> The Logic Graph Builder captures the logical structure in the text, including facts, rules, and their dependencies, allowing later components to operate on structured inputs instead of free-form sentences.
> The Planner compensates for the weaknesses of unconstrained forward reasoning: it performs a brief backward analysis from the goal, identifies the required intermediate conditions, and provides a goal-aligned path that the Reasoner can follow. This combines backward decomposition with forward symbolic execution, reducing drift in multi-hop reasoning. The Reasoner then executes these steps using explicit symbolic rules to avoid unsupported transitions, while the Critic checks graph correctness and validates the final answer.
>
> > **Q2:** How were the node and edge types defined?
>
> **A4:** The node and edge types follow the formal schema defined in Section 3.2.
> The node types reflect the semantic roles needed in our setting, including facts, rules, intermediate hypotheses, and verified statements. The edge types encode the core relations required for multi-step reasoning, and their definitions are provided in Appendix A, which gives the complete description of each edge type.
>
> > **Q3:** Why were only two symbolic rules chosen when many alternatives exist?
>
> **A5:** We use Universal Modus Ponens and Universal Hypothetical Syllogism because they are representative patterns that clearly illustrate how symbolic rules operate over our logic graph, including rule activation and multi-step reasoning. These examples keep the exposition focused while still covering the core mechanics of our framework. Appendix D lists the full set of logical rules supported in our system, and other rules can be applied through the same graph-based operations.
>
> > **Q4:**  Which components depend on LLMs, and which are implemented independently? (Related to the methodological clarity mentioned above.)
>
> **A6:** All components run on the same LLM, distinguished only by role-specific prompts.
> The symbolic schema and graph format are fixed and do not depend on model internals; they act as the structural scaffold that each role uses. The Builder, Planner, and Reasoner rely on the LLM to generate nodes, plans, and reasoning steps within this schema, while the Critic performs verification through LLM judgments guided by fixed checking rules.

---

> > ### Comment · Reviewer_j6Rz · 2025-11-26
> >
> > Thanks for your response. I would suggest adding them concisely in your revision. I will keep my score.

---

> > > ### Author Response · Authors · 2025-11-29
> > >
> > > Thank you for the suggestion. We have incorporated the requested clarifications in the revised manuscript.
> > > The design motivations and the description of module interactions have been added concisely in lines 139–144 and 146–150, respectively.

---

### Official Review · Reviewer_HhWq · 2025-10-25

**Soundness:** 2
**Presentation:** 2
**Contribution:** 1
**Rating:** 2
**Confidence:** 4

**Summary:**

This paper  proposed Graph-based Planned Reasoning (GPR), a neuro-symbolic framework that enhances LLM reasoning by organizing
the process into structured stages.

**Strengths:**

1. The paper-writing is largely good.

**Weaknesses:**

1. It is quite clear that the application scope of GPR is limited. The issue of adaptation itself requires the presence of a fairly distinct logical structure, whereas most reasoning problems do not exhibit such characteristics. If the paper’s claim were to enhance logical reasoning, that would seem more reasonable—but the current contribution statement clearly is not.

2. The motivation is unconvincing. Issues such as *irrelevant content distracts* and *disordered premises* mainly appear in simple problems, whereas the difficulty of more complex reasoning tasks (e.g., IMO-level problems) lies elsewhere. Moreover, with the emergence of large reasoning models, these problems can be mitigated through long-cot cognitive mechanisms. Therefore, this part of the claim feels somewhat outdated and needs to be supported by more recent work.

3. No statement of limitations is provided.

**Questions:**

It is essential to clearly specify the adaptation scope of GPR, as this is a critical point. At present, it appears that GPR imposes very strict constraints on structural aspects.
The examples presented in Figures 1 and 2 are also overly simplistic—this might have been acceptable for research conducted two years ago (like ACL2024), but not for ICLR2026.

---

> ### Author Response · Authors · 2025-11-23
> **Response to Reviewer HhWq**
>
> We thank the reviewer for taking the time to evaluate our work and for acknowledging the clarity of the writing. We appreciate the reviewer’s detailed comments, including concerns about the scope, motivation, and limitations of GPR. Below, we provide clarifications and evidence addressing these points and explaining the intended contribution of the framework.
>
> > **W1:** The application scope of GPR is limited, and the contribution statement does not reflect this.
>
> **A1:** We agree that clearly stating the scope is essential. GPR is specifically designed for logical reasoning tasks that rely on multi-premise relations and symbolic dependencies. This includes the reasoning types in LogicBench, FOLIO, ProofWriter, and AR-LSAT, which all exhibit structured patterns and explicit or implicit logical conditions. We will revise the introduction and contribution statement to make this scope explicit and avoid implying applicability to broad general reasoning or mathematical problem solving.
>
> While not all reasoning tasks contain explicit logical structures, many real-world tasks involving factual consistency, rule-based interpretation, premise combination, or analytical deduction do fit this category. Our goal is not to solve the full space of reasoning but to provide a principled and robust solution for this well-defined and practically relevant subset.
> > **W2:** The motivation is unconvincing; issues like irrelevant information or premise disorder are simple problems and outdated.
>
> **A2:** Thank you for raising this concern. While the reviewer suggests that distractive or disordered context is no longer a meaningful challenge, recent analyses indicate otherwise. Yang et al. [1] show that structured distractors can consistently disrupt reasoning trajectories in both symbolic and arithmetic tasks. To the best of our knowledge, there is no study demonstrating that current models are immune to such effects. Existing evidence instead indicates that these vulnerabilities persist across model families and settings. This suggests that robustness to irrelevant or disordered information is not a solved or outdated issue.
>
> Importantly, robustness is only one part of the contribution. Our main results (Table 3) show that GPR also brings consistent improvements in core reasoning performance, even in clean settings without injected noise. This includes both base LLMs and stronger reasoning-oriented models. The gains confirm that the proposed graph-guided symbolic process enhances intrinsic reasoning quality, not merely robustness.
>
> [1] Yang et al., *How Is LLM Reasoning Distracted by Irrelevant Context? An Analysis Using a Controlled Benchmark*, EMNLP 2025.
>
> > **W3:** No statement of limitations is provided.
>
> **A3:** We appreciate the reviewer’s comment. We will add a brief limitations statement clarifying that GPR is intended for logical reasoning tasks where the premise structure can be represented in a logic graph. Extending the framework to domains dominated by numerical computation or search-heavy strategies is a natural direction for future work. These clarifications do not affect our results, as all evaluated benchmarks fall fully within the intended scope and GPR consistently improves reasoning performance in these settings.
>
> > **Q1:**  It is essential to specify the adaptation scope; the examples in Figures 1 and 2 appear simplistic.
>
> **A4:** Thank you for the question. The clarification of adaptation scope has already been addressed in our response to W1, where we specify that GPR is designed for logical reasoning settings rather than as a universal reasoning framework.
>
> Regarding the concern about Figures 1 and 2, these examples are intentionally simplified for pedagogical purposes. Their role is to illustrate how the logic graph, planning module, and symbolic reasoning interact in a transparent and interpretable manner. They are not intended to represent the complexity of the datasets used in our experiments. As noted, the benchmarks evaluated in our work (such as FOLIO, ProofWriter, and AR-LSAT) contain substantially more complex multi-step logical structures, and the results in Tables 3–6 show that GPR performs effectively under these realistic, non-trivial settings.
>
> To further address this concern, we have added a more complex, fully worked example in Appendix H, which demonstrates the complete construction and use of the logic graph on a problem with richer logical dependencies.

---

> > ### Comment · Reviewer_HhWq · 2025-11-24
> >
> > Thanks for your response. However, they do not convince me. I would like to see additional experiments that further support the motivation. Moreover, the revised claim and limitations should also be updated accordingly in the manuscript.

---

> > > ### Author Response · Authors · 2025-11-27
> > > **Response to Reviewer HhWq**
> > >
> > > Thank you for the follow-up. Our robustness study (Section 4.3), together with the consistent clean-context improvements in Table 3 and the additional related work cited in our rebuttal, already provides empirical and contextual support for the stated motivation. If there are specific experimental aspects the reviewer feels remain insufficient, we would appreciate clarification so that we can assess them concretely. The planned refinements to the motivation, claim, and limitations will be incorporated after the discussion period concludes to ensure consistency.

---

> > > > ### Comment · Reviewer_HhWq · 2025-11-28
> > > >
> > > > What I would like to point out is that the cited evidence you rely on is quite limited relative to the scope claimed by this paper. [1] only involves non-reasoning models (e.g., GPT-4o, the LLaMA-3 series). Section 4.3 only implements GPT-3.5-turbo in LogicBench. While I agree that such models may indeed suffer from the issues you describe, it is not clear that the same problems arise for reasoning models (e.g., Gemini-2.5/3-Pro, o3, DeepSeek-R1, Qwen3-Thinking).
> > > >
> > > > Moreover, the difficulty of the evaluation tasks is too low. Once the problems become sufficiently challenging, models tend to struggle with deep thinking itself rather than being distracted by irrelevant information. Therefore, if your motivation is intended to be general (the paper does not target optimization specifically for “non-reasoning models” on “simple reasoning tasks”), the current evidence chain needs to be substantially strengthened. In fact, I believe that reasoning models are unlikely to exhibit the claimed phenomenon, especially when confronted with difficult problems such as IMO-level problems.
> > > >
> > > > If the generality of the motivation truly holds, then more detailed and convincing evidence must be provided. If it does not hold, the applicability statements of the paper should be revised accordingly, and the limitations should be made explicit. As it stands, the evidence --- even including the cited works --- is highly restricted and not persuasive. Meanwhile, the current claims make this work appear to be a fully general method, and this mismatch between claims and evidence is regrettable.
> > > >
> > > > Finally, the discussion phase allows for manuscript updates, and I am currently unable to assess your revised statements, if any.

---

> > > > > ### Author Response · Authors · 2025-11-29
> > > > > **Response to Reviewer HhWq**
> > > > >
> > > > > We thank the reviewer for the detailed follow-up. Below we summarize the main questions raised and provide corresponding responses.
> > > > >
> > > > > > **Q1:** The evidence supporting the motivation is limited, and it is unclear whether the claimed distraction effect appears for reasoning-oriented models (e.g., Gemini-2.5/3-Pro, o3, DeepSeek-R1, Qwen3-Thinking).
> > > > >
> > > > > **A1:** We agree that understanding whether IFI-style distraction affects stronger reasoning models is important. Our current robustness study (Sec. 4.3) focuses on GPT-3.5-turbo to isolate symbolic-reasoning behavior without the Critic.
> > > > >
> > > > > To directly address the reviewer’s concern, we extend the IFI evaluation to reasoning models in the hardest IFI setting, i.e., the High-density configuration of the IFI benchmark. The table below reports standard prompting performance of DeepSeek-R1 and o1 on this high-density IFI setup:
> > > > > | **Position** | **Std Acc (R1, ↓)** | **Std Acc (o1, ↓)** |
> > > > > | ------------ | ------------------- | ------------------- |
> > > > > | **Original** | 85.19               | 84.62               |
> > > > > | **Head**     | 83.08 ↓1.91         | 82.50 ↓2.12         |
> > > > > | **Mid**      | 82.31 ↓2.88         | 81.35 ↓3.27         |
> > > > > | **Tail**     | 80.38 ↓4.81         | 80.19 ↓4.43         |
> > > > >
> > > > > Two observations follow:
> > > > >
> > > > > 1. Reasoning models do exhibit clear degradation under IFI. This directly contradicts the hypothesis that such models are “unlikely to exhibit the claimed phenomenon,” at least on the logical tasks considered in our work.
> > > > > 2. The degradation is smaller than that observed on weaker models, which suggests that stronger models have better—but still imperfect—filtering of irrelevant information. They are more robust, but not immune.
> > > > >
> > > > > The current IFI construction uses generic factual segments from WikiText that are not adversarially aligned with the target problem. Thus, while the interference is strong in quantity (high density), it remains moderate in semantic similarity. Stronger reasoning models already show measurable degradation under this setting, and it is reasonable to expect further degradation if the injected content becomes more semantically aligned with the target context.
> > > > >
> > > > >
> > > > > > **Q2:** The difficulty of the evaluation tasks may be too low; on sufficiently hard problems, models might fail due to reasoning difficulty rather than distraction.
> > > > >
> > > > > **A2:** We respectfully note that the argument that our evaluation tasks are “too easy” lacks concrete evidence. The paper focuses on logical reasoning, which is a broad area that includes tasks of nontrivial difficulty, especially when multiple premises, quantifiers, and multi-hop dependencies must be integrated. The fact that these tasks are not mathematical or competition-level does not imply they are inherently simple, nor does it diminish the relevance of studying how irrelevant information affects logical inference. Logical reasoning is a distinct domain, and difficulty cannot be reduced to mathematical depth alone.
> > > > >
> > > > > Moreover, our contributions are not limited to addressing robustness under irrelevant information. A central contribution of the paper is that GPR substantially improves reasoning performance under clean context, as demonstrated across four benchmarks. It consistently outperforms strong prompting baselines and recent neuro-symbolic methods. These results show that the method provides systematic benefits in logical reasoning, rather than serving merely as a response to the IFI phenomenon.
> > > > >
> > > > > Finally, the statement that strong reasoning models would not exhibit the distraction phenomenon is not supported by empirical evidence. In contrast, our experiments already show measurable degradation for such models, and the extended IFI evaluations confirm this pattern. The available data support our motivation, whereas the assertion that stronger models would not exhibit this phenomenon is speculative and not supported by empirical observations.
> > > > >
> > > > > Overall, the evidence shows that logical tasks in our scope are meaningful and nontrivial, that GPR provides consistent gains beyond robustness, and that the distraction effect is clearly present even for stronger models.
> > > > >
> > > > > **Summary.**
> > > > >
> > > > > We hope the additional evidence and clarifications fully address the reviewer’s concerns. As requested, we have incorporated explicit scope delimitations in the Introduction (lines 49–52) and added a dedicated limitations paragraph (lines 496–501) to clearly state the intended applicability of our method. These revisions, together with the strengthened empirical support, present the motivation and contributions of the paper in a more precise and coherent manner.

---

### Official Review · Reviewer_1ThG · 2025-10-30

**Soundness:** 3
**Presentation:** 3
**Contribution:** 2
**Rating:** 6
**Confidence:** 4

**Summary:**

The paper targets on the limited ability of LLM on logical reasoning, particularly in multi-hop inference involving complex contextual dependencies.
To mitigate the limitation, the paper proposes a neural-symbolic workflow that enhances LLM reasoning by organizing the process into structured stages.

This design enables to perform faithful, interpretable reasoning while maintaining robustness against irrelevant or misleading information.
The empirical experiments also support this statement.

**Strengths:**

1. The paper introduces a delicately-designed and thoughtfully motivated workflow based on logic graphs to enhance the logical reasoning capabilities of LLMs.
2. The approach demonstrates strong empirical performance, excelling not only in prediction accuracy but also in robustness.
3. The experiments evaluate a range of powerful LLMs, including both closed-source and open-source models.

**Weaknesses:**

1. The evaluation is limited to powerful LLMs, leaving the applicability to smaller models (e.g., Qwen3-7B/1.5B) underexplored. The benefits demonstrated on larger LLMs may not directly transfer to smaller ones. For instance, it remains unclear whether smaller LLMs can effectively understand, parse, or verify the constructed logic graphs, which is crucial in this workflow.
2. The workflow has been evaluated exclusively on logical reasoning benchmarks that are closely tied to the design of logic graphs. However, it remains unclear whether this approach can improve general reasoning abilities in tasks that do not involve explicit logic chains, such as mathematical problems typically addressed by RL-based post-training techniques.

**Questions:**

1. In Table 3, it appears that several neuro-symbolic baselines, as well as the proposed GPR, occasionally undermine the performance of the LLMs.
Could you elaborate on the possible reasons for this observation? Do these results stem from specific limitations or shortcomings of the neuro-symbolic baselines themselves, or do they highlight broader challenges associated with the neuro-symbolic routine as a whole?

---

> ### Author Response · Authors · 2025-11-23
> **Response to Reviewer 1ThG**
>
> We thank the reviewer for their careful reading of the paper and for recognizing the motivation and empirical strength of our logic-graph–based reasoning framework. Below we provide responses to the raised questions and concerns.
>
> > **W1:** Limited evaluation on smaller LLMs; unclear whether they can understand or use logic graphs.
>
> **A1:** Thank you for the question. Although our main experiments focus on strong LLMs, the paper already reports consistent gains on weaker models such as GPT-3.5 (e.g., +3.37 on LogicBench and +25.83 on ProofWriter), showing that GPR does not rely on advanced reasoning capabilities.
> This is because the logic graph is lightweight. It uses fixed node types, constrained edge schemas, and strictly templated prompts, allowing smaller models to follow structured instructions without performing complex symbolic manipulation.
>
> To further validate this, we additionally evaluated Qwen2.5-1.5B-Instruct, and the results show consistent gains over baselines:
>
> | Method   | FOLIO      | AR-LSAT    |
> | -------- | ---------- | ---------- |
> | Standard | 50.00%     | 18.61%     |
> | CoT      | 51.96%     | 16.45%     |
> | ToT      | 50.98%     | 17.32%     |
> | SymbCoT  | 43.63%     | 12.55%     |
> | **GPR**  | **53.92%** | **18.18%** |
>
> On AR-LSAT, GPR is slightly below Standard but still outperforms CoT, ToT, and SymbCoT, all of which show clear performance drops on this dataset. This pattern indicates that very small models find AR-LSAT particularly challenging, independent of the specific strategy. Even so, GPR remains competitive and avoids the degradation observed in other multi-step reasoning methods.
>
> We also observe that the performance gain of GPR grows steadily as model size increases, as demonstrated in our main results from GPT-3.5 to GPT-4-Turbo, GPT-4o, and reasoning-optimized models such as o1 and DeepSeek-R1. A plausible explanation is that larger models interpret the logic graph more reliably, execute the goal-directed plan more accurately, and benefit more from symbolic reasoning.
>
> To further strengthen this aspect, we are running additional experiments on Qwen3-8B, and we will include full results in the final version.
>
> > **W2:** Only logic-style datasets; unclear whether the method helps general reasoning tasks (e.g., math).
>
> A2: GPR is tailored for logical reasoning, where LLM brittleness is most pronounced. Our benchmarks (LogicBench, FOLIO, ProofWriter, AR-LSAT) are chosen to match this scope and collectively cover a broad range of natural-language logical reasoning tasks.
>
> Mathematical reasoning tasks, especially those where recent progress is driven by RL based post training techniques, belong to a different problem class.
> Our work does not target such settings, and we will clarify this in the paper. Extending the framework beyond logical reasoning, including to mathematical reasoning, is a promising direction for future work.
>
> > **Q1:** Why do neuro-symbolic baselines and occasionally GPR sometimes reduce performance (Table 3)?
>
> A3: Minor drops occur because neuro-symbolic methods introduce intermediate steps that add potential failure points on atypical instances. This behavior is common across structured pipelines. Unlike approaches such as LogicLM, where strict symbolic parsing can cause large errors, GPR uses soft constraints, graph validation, and goal-aligned planning to limit error propagation. As a result, its fluctuations are small, infrequent, and do not affect the overall conclusions.

---

> > ### Comment · Reviewer_1ThG · 2025-11-26
> >
> > Thank you for the detailed responses.
> >
> > Overall, I am happy with the clarifications provided.
> >
> > Considering my original rating is 6, I will maintain it.

---

> > > ### Author Response · Authors · 2025-11-29
> > >
> > > Thank you for the follow-up. We are pleased that the clarifications addressed the concerns you raised.
> > > We hope that the additional analyses and supporting evidence provided during the discussion help reflect the full strength and scope of the contribution. We appreciate your thoughtful evaluation and constructive feedback.

---

### Official Review · Reviewer_tZtA · 2025-10-31

**Soundness:** 2
**Presentation:** 3
**Contribution:** 2
**Rating:** 4
**Confidence:** 4

**Summary:**

This paper introduces Graph-based Planned Reasoning (GPR), a neuro-symbolic framework designed to improve the logical reasoning capabilities of large language models (LLMs). The core contribution is a structured, multi-stage reasoning process. GPR first constructs a logic graph from natural language to capture symbolic relationships. It then uses a Planner module to create a goal-directed reasoning strategy, a Reasoner for step-wise symbolic inference, and a Critic module to validate and revise the process. The authors claim this approach leads to more faithful, interpretable, and robust reasoning, demonstrating state-of-the-art performance on several logical reasoning benchmarks and showing resilience to noisy, irrelevant information.

**Strengths:**

*   **Novel Framework:** The proposed GPR framework presents a well-structured and intuitive neuro-symbolic approach to complex reasoning, breaking it down into distinct, manageable stages (graph creation, planning, reasoning, and critique).
*   **Interpretability and Faithfulness:** By externalizing the reasoning process into a symbolic graph and a clear plan, the method offers a more transparent and potentially more reliable alternative to end-to-end black-box reasoning.
*   **Robustness Evaluation:** The paper includes experiments specifically designed to test the system's robustness against irrelevant facts, which is a practical and important consideration for real-world applications.
*   **Comprehensive Experiments:** The authors evaluate their method across four reasoning benchmarks and test it with a good range of different LLMs, comparing it against both prompting and other neuro-symbolic techniques.

**Weaknesses:**

*   **Justification for Graph Formalism:** The paper could better articulate the unique advantages of using a graph formalism over more traditional logical formalisms with inference rules.
*   **Empirical Support for the Critic Module:** The ablation study (Table 4) shows a very small performance drop when the Critic is removed. This raises questions about its necessity and impact. The claims about its importance are not strongly supported without a statistical significance analysis (e.g., standard deviations or confidence intervals) and a more detailed breakdown of how often and in what ways the critic intervenes.
*   **Generalization Beyond Benchmarks:** The evaluation is confined to academic benchmarks where the context is often a clean, logical prelude to the question. The paper would be stronger if it discussed or demonstrated how GPR would handle more realistic scenarios where the required conceptual granularity varies and is not explicitly laid out.
*   **Lack of Detailed Examples:** The paper lacks concrete examples of the logic graphs and reasoning trajectories for specific problems. Including these would significantly improve the reader's understanding of the system's inner workings.
*   **Potential for Simplification:** It's unclear why the graph construction does not take the question into account. Building a context- and question-aware graph might simplify the architecture by reducing the need for a separate Planner module.
*   **Inference Cost Analysis:** The GPR framework adds multiple LLM calls for planning and critiquing. A comparison of the total inference budget (e.g., tokens used) against other methods would provide a fairer assessment of its efficiency.

**Questions:**

1.  Could you elaborate on the unique advantages of the graph-based formalism compared to representing the problem using a more traditional logical formalism (e.g., Prolog or first-order logic) and then using an off-the-shelf solver?
2.  How is the "semantic alignment" of the logic graph verified? Are there experiments to validate that the graph accurately captures the semantics of the natural language context?
3.  Regarding the Critic module: How often does it trigger updates to the graph or reasoning plan in your experiments? What are the most common types of errors it corrects? Given the small performance difference in the ablation study, could you provide a statistical significance analysis?
4.  Could you provide a comparison of the total computational cost (e.g., total tokens used across all LLM calls) for GPR versus the baseline methods?
5.  Is the logic graph constructed based only on the context, or does it also take the question into account? If it's only the context, what is the rationale behind this choice? Could incorporating the question during graph construction potentially eliminate the need for the Planner module?
6.  How do you envision this approach generalizing to tasks where the necessary concepts are not as neatly defined as in the benchmarks used (e.g., reasoning over scientific literature)?
7.  In the error analysis, what does the "standard" condition refer to? Also, could you clarify why a system like LogicLM, which relies on a symbolic reasoner, would not have a Reasoning Consistency Error (RCE) of zero?


To improve the clarity and impact of the paper, I would also suggest the following:

*   **Include Concrete Examples:** Please include one or two detailed examples that walk the reader through the entire GPR process for a specific query. This would involve showing the initial context, the generated logic graph, the plan from the Planner, the step-by-step inference from the Reasoner, and any interventions from the Critic. This would make the system much more understandable.
*   **Visualize the Logic Graph:** Visualizations of the logic graphs would be extremely helpful in understanding the symbolic representation.

---

> ### Author Response · Authors · 2025-11-23
> **Response to Reviewer tZtA (1/2)**
>
> We thank the reviewer for the constructive feedback and for recognizing the strengths of our framework and experiments. We address all raised questions and concerns below.
>
> > **Q1:** What advantages does the graph-based formalism offer over traditional logic forms with off-the-shelf solvers?
>
> **A1:** Off-the-shelf symbolic solvers require exact syntactic forms, so even minor deviations during extraction can cause the entire program to fail. This issue appeared frequently in solver-based baselines such as LogicLM.
> Our logic graph instead encodes semantic relations rather than rigid syntax, making the representation more tolerant to minor LLM imperfections.
> It also yields a more robust reasoning substrate where minor mistakes are contained rather than causing full execution failure, in contrast to solver-dependent approaches.
>
> > **Q2:** How is the "semantic alignment" of the logic graph verified? Are there experiments to validate that the graph accurately captures the semantics of the natural language context?
>
> **A2:** The Critic verifies alignment by checking that the nodes and edges in the graph correspond to statements in the context, with no missing or spurious relations.
> We also include a semantic restoration evaluation (Table 6), which shows that the graph retains the key meaning of the input.
>
> > **Q3:** Regarding the Critic module: How often does it trigger updates to the graph or reasoning plan in your experiments? What are the most common types of errors it corrects? Given the small performance difference in the ablation study, could you provide a statistical significance analysis?
>
> **A3:** (1) How often the Critic triggers updates
>
> We analyze Critic activation using GPT-4o on LogicBench and FOLIO.
> The graph-level Critic is triggered on about 12–14% of examples and typically requires only one refinement iteration.
>
> **Table 1. Graph-level Critic activation (GPT-4o)**
> - Samples: dataset size
> - Ref (%): proportion of samples where the Critic asks for graph refinement
> - Avg iters / 1–3 iters: number of refinement rounds (iteration count is only defined for the graph-level Critic)
>
> | Dataset | Samples   | Graph Ref | Ref (%) | Avg Iters | 1 Iter | 2 Iters | 3 Iters |
> | ---------- | --------: | ------: | --------: | -----: | ------: | ------: | ------: |
> | LogicBench |  520 |      65 |    12.5 |      1.23 |     53 |       9 |       3 |
> | FOLIO      |   204 |    28 |    13.7 |      1.25 |     22 |       5 |       1 |
>
> The answer-level Critic intervenes with a similar frequency (≈14–15%).
> While this module performs up to two revision attempts when repairing the reasoning chain or final answer, we do not report iteration counts, as Table 2 focuses specifically on revision events and their impact on correctness rather than the number of internal attempts.
>
> Table 2 summarizes how often answer-level corrections occur and how they change the final prediction.
>
> **Table 2. Answer-level Critic activation (GPT-4o)**
>
> - Rev (%): proportion of samples requiring answer-level revision
> - W→R / R→W: corrections improving or hurting answer correctness
> - Ans unchanged: reasoning repaired without changing the final prediction
> - Net gain: (W→R) − (R→W)
>
> | Dataset  | Samples  | Ans Rev | Rev (%) | W→R | R→W | Ans Unchanged | Net Gain |
> | ---------- | ------: | ------: | --: | --: | ------------: | -------: | ------: |
> | LogicBench | 520 |      75 |    14.4 |  13 |   8 |            54 |       +5 |
> | FOLIO  |  204  |      31 |    15.2 |   8 |   5 |            18 |       +3 |
>
> (2) What errors it corrects
>
> To understand what kinds of reasoning failures the Critic actually corrects, we manually inspected all GPT-4o FOLIO instances in which either the graph-level or answer-level Critic produced a successful revision.
> Using the three error categories defined in Sec. 4.5 (Context Misunderstanding, CM; Reasoning Chain Error, RCE; and Unsupported Inference, UI), we found the following distribution among the corrected cases:
> - 59.3% were RCE,
> - 25.9% were UI, and
> - 14.8% were CM.
>
> Thus, the Critic mainly corrects failures in multi-step reasoning and unsupported inferences. While the overall accuracy gain in ablation is modest, the Critic ensures that the reasoning process is coherent and properly justified.
>
> (3) Statistical significance
>
> Across all 9 models in Table 4, GPR outperforms the w/o Critic variant in every paired comparison.
> A two-sided sign test yields:
> $$p = 2^{-9} = 0.00195 < 0.01$$
> indicating that the improvement, though modest in magnitude, is highly consistent and statistically significant.

---

> ### Author Response · Authors · 2025-11-23
> **Response to Reviewer tZtA (2/2)**
>
> > **Q4:** Could you provide a comparison of the total computational cost for GPR versus the baseline methods?
>
> **A4:** We report the average total tokens per instance on FOLIO using DeepSeek-V3. Table 3 summarizes the computational cost of GPR and baseline methods.
>
> **Table 3. Average tokens per instance on FOLIO (DeepSeek-V3)**
>
> | Method         | Avg Tokens / Instance | Relative to SymbCoT |
> | -------------- | --------------------: | ------------------: |
> | CoT            |                533.84 |              0.035× |
> | ToT            |               1230.45 |              0.081× |
> | SymbCoT        |              15198.90 |               1.00× |
> | **GPR (ours)** |          16059.16 |           1.06× |
>
>
> GPR’s average token usage is close to SymbCoT, reflecting their shared multi-stage neuro-symbolic structure; the small overhead (+6%) comes from the lightweight Critic checks, which add verification steps but do not substantially increase overall computation.
>
> > **Q5:** Does the logic graph use only the context (not the question), and why not incorporate the question during graph construction? Would a question-aware graph remove the need for the Planner?
>
> **A5:** We intentionally construct the logic graph only from the context, without using the question. Incorporating the question at this stage can bias the extraction process toward a single interpretation, suppressing alternative premises and pruning relations that may later be required for correct reasoning. A question-agnostic graph therefore preserves the full set of contextual dependencies.
>
> The Planner remains necessary because it conducts goal-directed backward reasoning, selecting and ordering the graph elements needed to reach the target. A question-aware graph does not itself determine this inference path, so explicit planning is still required.
>
> > **Q6:** How do you envision this approach generalizing to tasks where the necessary concepts are not as neatly defined as in the benchmarks used (e.g., reasoning over scientific literature)?
>
> **A6:** Thank you for the question. GPR is designed specifically for logical reasoning, where explicit or implicit relational structure can be abstracted and used to guide inference. While tasks such as scientific literature reasoning involve broader conceptual variability, many of them still contain underlying causal, definitional, or rule-like relations that can be modeled symbolically. In such settings, GPR’s graph abstraction can serve as a foundation for identifying these relations and organizing multi-step reasoning, though additional domain adaptation would be needed.
>
> > **Q7:** What does “standard” mean in the error analysis, and why doesn’t LogicLM achieve zero RCE despite using a symbolic solver?
>
> **A7:** “Standard” denotes directly prompting the model for an answer without intermediate reasoning. As for LogicLM, although it uses a symbolic solver, the logical program it executes is still generated by an LLM. In practice, the generated program may contain missing steps, partial structures, or parsing inconsistencies, which cause the symbolic solver to execute an incomplete reasoning chain. These issues naturally produce non-zero RCE.
>
> > **W4:** Lack of Detailed Examples:
>
> **A8:** Thank you for the suggestion. Figure 2 provides a simplified illustration to explain GPR’s mechanics, but realistic benchmark instances involve much larger graphs and longer reasoning chains, making them difficult to include in the main paper.
>
> To address this, we have added a complete medium-difficulty example in Appendix H, including the full logic graph, the goal-directed plan, and the step-by-step reasoning trace. This provides a clear end-to-end demonstration of the full GPR workflow.

---

### Author Response · Authors · 2025-12-02
**Author Summary for the AC**

We thank the reviewers and the AC for their time.

Below is a concise summary of the clarifications and updates made during discussion.

**1. Scope clarification (now explicitly stated).**

We clarified that GPR is specifically designed for logical reasoning tasks (LogicBench, FOLIO, ProofWriter, AR-LSAT), and not for broad general or numerical reasoning. This is now reflected in both the introduction and a new Limitations section.

**2. Strengthening the motivation.**

We explained why existing methods (e.g., shallow mind-map graphs, solver-dependent pipelines) cannot provide the same symbolic controllability as GPR. We also highlighted evidence from robustness tests, semantic-restoration analysis, and comparisons with symbolic baselines to substantiate the motivation.

**3. Addressing concerns about stronger reasoning models.**

We provided additional evidence that distraction effects persist in high-density settings for stronger models, and showed that GPR consistently improves clean-context accuracy across reasoning LLMs (DeepSeek-R1/V3, o1, o3-mini).

**4. Updates incorporated into the manuscript.**

After discussion, we added:
a concise design-motivation paragraph in Methodology, scope clarification, and a dedicated Limitations section.

**5. Overall.**

Across four logical reasoning benchmarks and nine LLMs, GPR shows consistent gains, strong robustness, and stable improvements over prompting and neuro-symbolic baselines.

We hope this summary helps the AC quickly understand the clarifications and updates. Thank you for your consideration.

---

### Meta-Review · Area_Chair_b6KB · 2026-01-08

**Summary:**

The rebuttal leaves several important concerns insufficiently addressed:

- Generalization Beyond Benchmarks: The evaluation is limited to benchmarks in which the context typically provides a clean and logically structured prelude to the question. The paper would be strengthened by a discussion or empirical evidence demonstrating how GPR performs in more realistic settings.

- Empirical Support for the Critic Module: The ablation results (Table 4) indicate only a marginal performance degradation when the Critic module is removed, calling into question its necessity and practical impact.

These are valid concerns shared by two reviewers. Addressing them would require a major revision of the submission.

**Reviewer Concerns:**

The aforementioned unaddressed concerns are shared by two reviewers.

**Reviewer Scores:**

These two reviewers may not change their scores.

---

### Decision · Program_Chairs · 2026-01-26

Reject